# MaXM: Towards Multilingual Visual Question Answering

**Soravit Changpinyo, Linting Xue, Michal Yarom, Ashish V. Thapliyal**
**Idan Szpektor**, **Julien Amelot**, **Xi Chen**, **Radu Soricut**
Google Research
https://github.com/google-research-datasets/maxm

## Abstract

Visual Question Answering (VQA) has been primarily studied through the lens of the English language. Yet, tackling VQA in other languages in the same manner would require a considerable amount of resources. In this paper, we propose scalable solutions to multilingual visual question answering (mVQA), on both data and modeling fronts. We first propose a translation-based framework to mVQA data generation that requires much less human annotation efforts than the conventional approach of directly collection questions and answers. Then, we apply our framework to the multilingual captions in the Crossmodal-3600 dataset and develop an efficient annotation protocol to create MaXM, a test-only VQA benchmark in 7 diverse languages. Finally, we develop a simple, lightweight, and effective approach as well as benchmark state-of-the-art English and multilingual VQA models. We hope that our benchmark encourages further research on mVQA.

## 1 Introduction

Visual Question Answering (VQA), the task of answering visual questions grounded in images, is key to human-machine interaction in the visual world. In particular, the natural language interface in VQA makes it easy for lay people to express their needs and benefit from its applications, including accessibility, education, and search. Yet, VQA advances were mostly focused on English, therefore only applied to a privileged subset of human populations.

Arguably, the English language has dominated the field mainly because of the availability of English VQA benchmarks. These benchmarks are diverse, from general VQA (Zhu et al., 2016; Kafle and Kanan, 2017; Krishna et al., 2017; Antol et al., 2015; Goyal et al., 2017; Changpinyo et al., 2022), robust VQA (Agrawal et al., 2018), compositional visual reasoning (Hudson and Manning, 2019), for

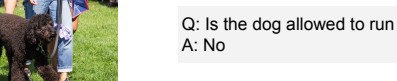
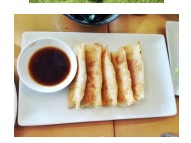
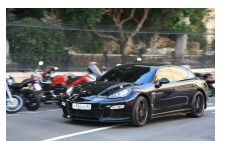
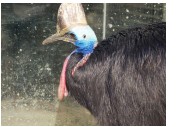
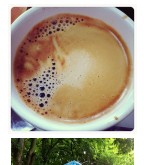
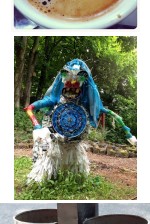
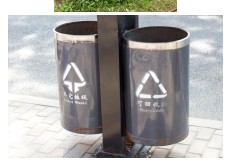

Figure 1: **Multilingual VQA Data in 7 languages.** The data is automatically generated from multilingual captions and then verified and adjusted by humans. From top to bottom: English (en), French (fr), Hindi (hi), Hebrew (iw), Romanian (ro), Thai (th), and Chinese (zh).

the blind and the visually-impaired (Gurari et al., 2018), scene-text understanding (Singh et al., 2019; Biten et al., 2019), to VQA that requires external, commonsense, or world knowledge (Marino et al., 2019; Zellers et al., 2019; Schwenk et al., 2022). These benchmarks require considerable amount of resources to create, mostly by employing human annotators to laboriously collect and verify

the questions and the answers for each image.

To extend VQA to all languages in the world, we must make data creation more automatic. Building on recent work on automatic data creation for English VQA from captions (Changpinyo et al., 2022), in this paper we propose a translation-based framework for multilingual visual question answering (mVQA) data creation. Our framework automates much of the task of generating questions and answers, thus providing a scalable path to mVQA.

We apply our framework to the generation of question-answer pairs from the multilingual captions of the recently-proposed Crossmodal-3600 dataset (XM3600) (Thapliyal et al., 2022). Combined with an efficient human annotation protocol, we construct MAVERICS-XM3600 (MaXM), a test benchmark for mVQA in 7 languages (see examples in Fig. 1).

Finally, we use this novel benchmark to drive progress in mVQA modeling and measure where we stand. We leverage advances in image modeling and multilingual modeling: ViT (Dosovitskiy et al., 2021) and mT5 (Xue et al., 2021) and propose a unified, extensible, open-ended mVQA model, called Simple MPT, which is competitive to state-of-the-art English VQA models that we adapt to apply in the mVQA setting (OFA (Wang et al., 2022b) and BLIP2 (Li et al., 2023)). Overall, there exists a large room for improvement.

In summary, our main contributions are (i) a scalable translation-based framework for mVQA data generation based on captions (Sect. 3); (ii) an efficient annotation protocol, deriving a novel test benchmark called MAVERICS-XM3600 (MaXM) in 7 diverse languages: English, French, Hindi, Hebrew, Romanian, Thai and Chinese (Sect. 4);Michal: Beer, why is this dataset better than xGQA? (iii) simple and lightweight mVQA modeling (Sect. 5.2, Sect. C) with strong performance; (iv) benchmarking (adaptations of) the state-of-the-art VQA models on MaXM (Sect. 5.3).

## 2  Related Work

### 2.1  VQA and Multilingual Multimodal Benchmarks

English has been the primary language in which vision-and-language researchers study the VQA task, driven by the availability of data and benchmarks (Zhu et al., 2016; Kafle and Kanan, 2017; Krishna et al., 2017; Antol et al., 2015; Goyal

et al., 2017; Agrawal et al., 2018; Gurari et al., 2018; Marino et al., 2019; Singh et al., 2019; Biten et al., 2019; Sheng et al., 2021; Li et al., 2021; Changpinyo et al., 2022). The only exception is xGQA (Pfeiffer et al., 2022), an extension of the English GQA dataset (Hudson and Manning, 2019). xGQA consists of human translations of the 12,578 English questions from 398 images in the balanced testdev split of GQA in 8 typologically diverse languages: English, German, Portuguese, Russian, Indonesian, Bengali, Korean, and Chinese. Besides the differences in the languages considered, our proposed approach to mVQA data creation complements xGQA (see Sect. 4.4).

Beyond mVQA, training and evaluation data for multilingual multimodal models is limited. For a review of previous work, we refer the reader to the Image-Grounded Language Understanding Evaluation (IGLUE) benchmark (Bugliarello et al., 2022), where xGQA is a part of. In general, early attempts often focus on Chinese (Li et al., 2019; Wang et al., 2019), Japanese (Yoshikawa et al., 2017; Aggarwal and Kale, 2020) and several Indo-European languages (e.g., German, French, and Czech) (Elliott et al., 2016, 2017; Barrault et al., 2018). However, there is a recent effort toward a wider variety of both languages and tasks. Examples include image retrieval (Aggarwal and Kale, 2020) (also Russian, Korean, Turkish), visual natural language inference (Bugliarello et al., 2022) (also Arabic), multilingual visual reasoning (Liu et al., 2021) (also Indonesian, Swahili, Tamil, Turkish), and vision-and-language navigation (Ku et al., 2020) (also Hindi, Telugu). Notably, Wikipedia Image Text (WIT) (Srinivasan et al., 2021) provides a large-scale image-text dataset in 108 languages, automatically collected form Wikipedia, and Crossmodal-3600 (XM3600) (Thapliyal et al., 2022) provides human-curated test-only image captions in 36 languages. Our work builds on top of XM3600, and the 7 languages that we consider are typologically, genealogically, and geographically diverse.

### 2.2  VQA Data Creation

Previous work on VQA data creation relies heavily on humans to create questions and answers (Zhu et al., 2016; Krishna et al., 2017; Goyal et al., 2017; Gurari et al., 2018; Marino et al., 2019). Some works attempt to automate this process. CLEVR (Johnson et al., 2017a) uses a template-

based approach, but it is based on synthetic images for which ground-truth annotations are available. GQA (Hudson and Manning, 2019) follows a similar approach but instead starts from Visual Genome scene graphs (Krishna et al., 2017), which themselves require large annotation efforts.

More relevant are works that rewrite image captions or video transcripts as question-answer pairs. COCOQA (Ren et al., 2015) uses a template-based approach that can only generate questions with one-word answers. WeaQA (Banerjee et al., 2021) improves upon this with semantic role labeling, paraphrasing, and backtranslation. Recently, Changpinyo et al. (2022) and Yang et al. (2021) leverage T5 (Raffel et al., 2020) fine-tuned on question answering datasets, generating large-scale VQA datasets for images and videos, respectively. Our approach to mVQA data creation leverages $VQ^2A$, the approach in (Changpinyo et al., 2022) (Sect. 3.1). To the best of our knowledge, besides xGQA, no other prior work on VQA data generation considered languages beyond English.

## 3 Multilingual VQA Data Creation

Like in many other machine learning tasks, the main bottleneck to mVQA is obtaining high-quality labeled data. The most popular data collection framework to English VQA is to ask a set of human annotators to come up with visual questions, and another set of annotator to answer them (Sect. 2.2). To scale VQA to all languages, we argue that mVQA data creation must significantly reduce its use of human annotation. To this end, we study the extension of an automatic English VQA data creation method called **V**isual **Q**uestion Generation with **Q**uestion **A**nswering validation, or $VQ^2A$ (Changpinyo et al., 2022) for the purpose of mVQA data creation.

### 3.1 Background: $VQ^2A$

The $VQ^2A$ approach leverages aligned image-text data sources that are available at scale (Ordonez et al., 2011; Chen et al., 2015; Sharma et al., 2018; Pont-Tuset et al., 2020; Changpinyo et al., 2021; Desai et al., 2021; Schuhmann et al., 2021) and beyond English (Srinivasan et al., 2021; Gu et al., 2022). It rewrites a declarative image caption into multiple interrogative question-answer pairs via three steps: (i) *Candidate Answer Extraction* extracts candidate answers based on syntactic and semantic analysis of an input caption, (ii) *Question*

*Generation* generates candidate questions for each candidate answer, (iii) *Answer Validation* filters candidate questions that do not pass a consistency check that involves automatically answering each question from the caption and comparing this answer to the original extracted answer (Alberti et al., 2019; Honovich et al., 2021).

Each step in $VQ^2A$ is optimized for English; Step (i) uses English spaCy and both Step (ii) and Step (iii) leverage high-capacity English-pre-trained T5 models fine-tuned on English question answering datasets (Rajpurkar et al., 2016, 2018; Kwiatkowski et al., 2019).

### 3.2 Translation-based $VQ^2A$ (TransVQ$^2$A)

Inspired by $VQ^2A$, our goal is to generate mVQA data at scale, leveraging multilingual image captions. Multilingualizing each step in $VQ^2A$ can be non-trivial and resource-intensive due to the heavy reliance of English tools, models, and data (Sect. 3.1). To alleviate this, we propose a translation-based extension of $VQ^2A$.

Given an input caption $c$ in any language, and a target language $\langle$lang$\rangle$, we want to generate question-answer pairs in $\langle$lang$\rangle$. We propose Translation-based $VQ^2A$ (TransVQ$^2$A), as follows: ***Step 1** Caption Translation*: Automatically translate a non-English caption $c$ to English $c_e$. ***Step 2** Apply* $VQ^2A$: Generate a set of English question-answer pairs $\{q_e, a_e\}$ from $c_e$. ***Step 3** Question-Answer Translation*: Automatically translate all $(q_e, a_e)$ pairs to $\langle$lang$\rangle$ $(q, a)$. ***Step 4** Validation*: Filter $(q, a)$ pairs[1] in which $a$ does not appear in the original caption $c$, back-translating $a$ to $c$'s language if necessary. The upper part of Fig. 2 exemplifies TransVQ$^2$A using a Chinese caption from Crossmodal-3600 (Thapliyal et al., 2022).

We highlight that the approach we have described so far is fully automatic and applicable to a huge set of languages that are supported by automatic translation. We note that the final validation is important due errors that could pile up during translation steps. This is especially acute in Step 3, since translating answers is harder due to the lack of disambiguating context in the short answers. We also note that TransVQ$^2$A can generate question/answer pairs in the target $\langle$lang$\rangle$ from any caption. The output quality depends on the translation quality, e.g. the back-translation in step 4 from $\langle$lang$\rangle$ to $c$'s language. We use out-of-the-box

---

[1]Excluding answers to boolean questions.

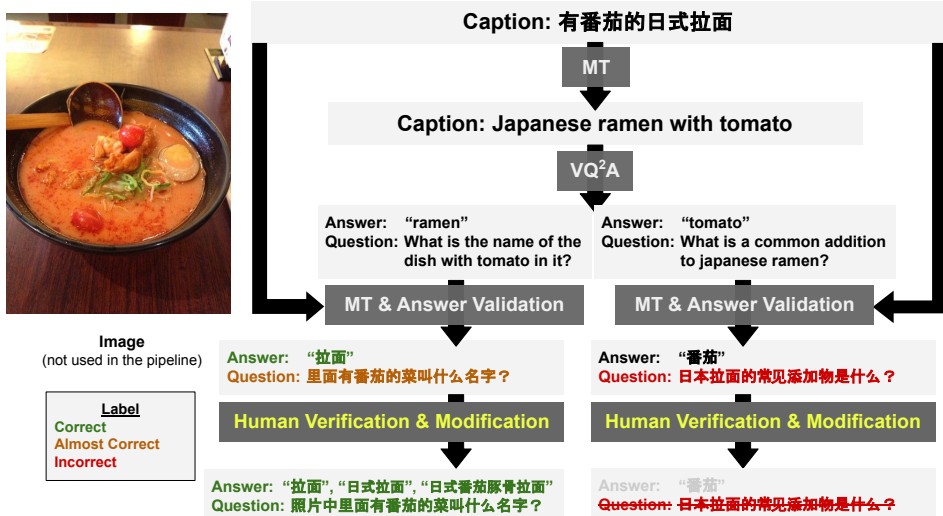

Figure 2: **Our approach to multilingual VQA data generation**, which is easy to scale, highly automatic and only requiring humans to modify "Almost Correct" questions or correct/expand answers (left) or filter out "Incorrect" questions(right). MT is short for automatic machine translation.

translation tools in this work, and leave the exploration of better translation tailored for $\mathrm{TransVQ^2A}$ for future work.

In Sect. 4 we employ human annotators to further clean and expand the generated data to create a high quality test benchmark.

### 3.3 Direct Question Generation (DirectQG)

One drawback of $\mathrm{TransVQ^2A}$ is the low coverage of particular types of answers, such as "no". This is because the captions generally do not indicate the absence of objects or properties (e.g., *"There is no dog"*, *"The dog is not white"*). To mitigate this bias, we train a multilingual question generator that takes in an answer and a caption in a target language and generates relevant questions in the same language. We use the model to generate questions for "yes", "no", or "none" as answers in each target language, as a complement to $\mathrm{TransVQ^2A}$.

Concretely, we fine-tuned mT5-XXL (Xue et al., 2021) on large-scale translated COCO Captions (Chen et al., 2015) and its corresponding VQA data $\mathrm{VQ^2A}$-COCO (Changpinyo et al., 2022). For validation, we used the subset of generated multilingual VQA data in Sect. 3.2, with ~300 golden examples for each language. The best checkpoint was selected based on ROUGE-L scores.

## 4  MaXM: Multilingual VQA Benchmark

In this section, we leverage the approach we presented in Sect. 3 for creating a multilingual VQA test-only benchmark. We next describe our data sources, how candidate data was generated, human

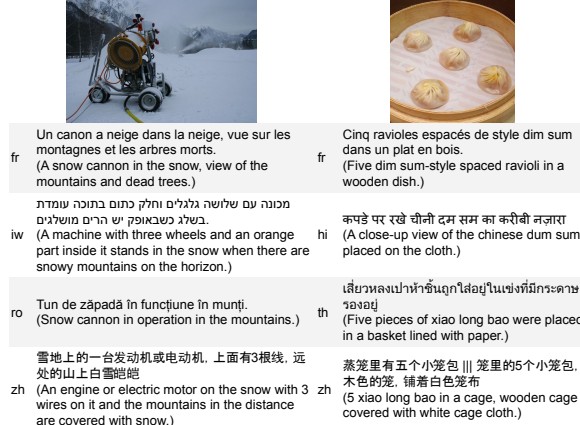

Figure 3: **The diversity of multilingual captions in** XM3600. We show the captions (their English translations) from 4 languages for the images of a snow cannon (left) and xiao long bao (right).

annotation protocol, and an analysis and a discussion of our benchmark. Following the naming convention in (Changpinyo et al., 2022), we call our benchmark MAVERICS-XM3600, or MaXM in short. We will release MaXM to foster research on mVQA.

### 4.1 Data Sources

**Language Selection**. We chose 7 languages that are 1) typologically diverse, 2) genealogically diverse, and 3) geographically diverse: English (en), French (fr), Hindi (hi), Hebrew (iw), Romanian (ro), Thai (th), and Chinese (zh).

**Image and Caption Selection**. We chose a subset of the images in Crossmodal-3600

|                            | en      | fr      | hi      | iw      | ro      | th      | zh      |
| -------------------------- | ------- | ------- | ------- | ------- | ------- | ------- | ------- |
| Captions                   | 7200    | 8562    | 8503    | 7200    | 7123    | 7200    | 7174    |
| English QAs                | 373248  | 499900  | 520080  | 544268  | 516604  | 415180  | 524252  |
| Validated English QAs      | 264930  | 343621  | 346948  | 375629  | 346887  | 286024  | 362304  |
|                            | (71.0%) | (68.7%) | (66.7%) | (69.0%) | (67.1%) | (68.9%) | (69.1%) |
| Validated Multilingual QAs | 264724  | 122644  | 153465  | 128613  | 121221  | 95531   | 182095  |
|                            | (99.92%)| (33.85%)| (53.67%)| (37.08%)| (32.27%)| (27.53%)| (52.99%)|

Table 1: *Number of Instances (% of Previous Stage)* of automatically-generated question-answer (QA) pairs based on Crossmodal-3600 captions. Validated English pairs are w.r.t the QG-QA consistency filter. Validated multilingual pairs are w.r.t the caption-answer consistency filter.

| Label          | Question                                                             | Answer                                        |
| -------------- | ------------------------------------------------------------------- | --------------------------------------------- |
| *Correct*        | Makes sense AND is relevant to the image.                          | Satisfies the question's intent wrt the image.|
| *Almost Correct* | Correct but its surface form can be improved (syntactic errors or awkward/uncommon usages.) | |
| *Incorrect*      | NOT Correct.                                                       |                                               |

Table 2: Definition of *Correct*, *Almost Correct* and *Incorrect* labels for questions and answers in our annotation protocol.

(XM3600) (Thapliyal et al., 2022), in which high-quality multilingual image captions are available. For each language, 100 validation and test images of Open Images (Krasin et al., 2017; Kuznetsova et al., 2020) that were taken in the region(s) in which those languages were spoken were selected.

Our image selection criteria cover a wide range of visual concepts in different cultural contexts, making the constructed VQA examples diverse and specific to the languages of the captions related to each image. For example, in Fig. 3, unlike French and Romanian speakers, Hebrew and Thai speakers are less likely to know what a snow cannon is. On the other hand, Thai and Chinese speakers are more likely to understand what xiao long bao is, whereas in French or Hindi it could be referred to as dim-sum ravioli or Chinese dim sum.

Another benefit of XM3600 is that the Open Images images are out-of-domain with respect to most widely-used VQA benchmarks (Ren et al., 2015; Antol et al., 2015; Zhu et al., 2016; Krishna et al., 2017; Goyal et al., 2017; Agrawal et al., 2018; Hudson and Manning, 2019; Marino et al., 2019), which are often based on MS-COCO images (Lin et al., 2014).

## 4.2 Large-Scale mVQA Data Creation

We apply our approach described in Sect. 3 to the XM3600 captions to generate a large number of question-answer pairs for each language.

**TransVQ$^2$A**. Table 1 reports the number of question-answer pairs at different stages in our pipeline. Overall, we are able to generate a large number of question-answer pairs in all languages. We found that, across languages, approximately 30% of (translated) English question-answer pairs are filtered out due to VQ$^2$A validation. In contrast, different percentages of translated answers across languages are filtered out based on the caption-answer consistency validation. A main reason for this is the quality of question-answer translation. For instance, 68% of questions with "alb" (masculine "white" in Romanian) are filtered out because they are not translated to the correct feminine form "alba" w.r.t the corresponding object in the question.

**DirectQG**. We augment the TransVQ$^2$A questions with additional candidate questions generated by TransVQ$^2$A (Sect. 3.3), using the XM3600 captions paired with "yes", "no", or "none" in their corresponding language as input.

## 4.3 Human Annotation

We employed native speakers for each of the selected 7 languages to annotate and create our benchmark. We designed an annotation protocol to balance efficiency and accuracy. In particular, we keep human-in-the-loop brief and only when an automated model straggles in a task, e.g., correcting translation artifacts, expanding answers, identifying sensitive questions. Furthermore, our protocol promotes quick discarding of examples, when the question does not make sense. We provide more details next and also in Appendix B.

**Question and Answer Validation**. We define a 3-way rating system of *Correct*, *Almost Correct* and *Incorrect* for both the questions and the answers (see Table 2). *Correct* questions are kept unchanged, *Almost Correct* questions are manually rewritten, and *Incorrect* questions are discarded. Given *Correct* and *Almost Correct* questions, an annotator rates the answer and corrects it in the cases of both *Almost Correct* and *Incorrect*.

Table 3 reports label distribution for questions and answers randomly-sampled from those gen-

| | en | fr | hi | iw | ro | th | zh |
|---|---|---|---|---|---|---|---|
| # of questions evaluated | 377 | 389 | 400 | 365 | 440 | 401 | 391 |
| % Correct | 62.6% | 65.8% | 66.5% | 61.6% | 59.1% | 47.4% | 51.9% |
| % Almost Correct | 17.5% | 12.3% | 9.0% | 22.5% | 9.3% | 27.9% | 25.1% |
| % Incorrect | 19.9% | 21.9% | 24.50% | 15.9% | 31.6% | 24.7% | 23.02% |
| # of answers evaluated | 302 | 304 | 302 | 307 | 301 | 302 | 301 |
| % Correct | 66.2% | 72.0% | 77.8% | 73.9% | 76.2% | 82.2% | 81.9% |
| % Almost Correct | 26.5% | 24.0% | 17.9% | 24.1% | 16.9% | 7.9% | 9.2% |
| % Incorrect | 7.3% | 3.9% | 4.3% | 2.0% | 7.0% | 9.9% | 8.9% |

Table 3: **Human evaluation** of the generated questions and answers.

| Question Prefix | Percentage | | | | | | |
|---|---|---|---|---|---|---|---|
| | en | fr | hi | iw | ro | th | zh |
| *"is"* | 22.8 | 21.2 | 21.4 | 17.8 | 19.5 | 16.2 | 20.2 |
| *"what is"* | 15.8 | 11.3 | 16.0 | 15.2 | 13.5 | 11.6 | 13.7 |
| *"how many"* | 15.1 | 11.3 | 13.9 | 10.2 | 14.1 | 15.2 | 12.1 |
| *"where"* | 6.7 | 9.2 | 7.5 | 8.6 | 6.6 | 9.3 | 5.9 |
| *"what kind"* | 6.0 | 7.2 | 1.4 | 3.8 | 6.6 | 4.0 | 3.6 |
| *"what are"* | 3.4 | 1.4 | 1.0 | 2.2 | 2.4 | 2.3 | 2.3 |
| *"who"* | 3.4 | 3.1 | 9.2 | 2.5 | 1.5 | 2.6 | 2.3 |
| *"are"* | 3.4 | 0.7 | 4.1 | 3.8 | 3.6 | 3.3 | 2.0 |
| *"what color"* | 3.4 | 7.2 | 5.1 | 9.2 | 6.9 | 8.6 | 7.5 |
| *"a"* | 3.0 | 3.1 | 1.4 | 3.5 | 2.1 | 3.0 | 2.9 |
| *"what type"* | 2.7 | 2.4 | 0.3 | 0.3 | 2.1 | 1.0 | 4.6 |
| *"what was"* | 1.0 | 0.3 | 0.0 | 0.6 | 1.2 | 3.3 | 0.7 |
| *"do"* | 0.7 | 0.3 | 0.0 | 0.3 | 0.6 | 0.7 | 2.3 |
| *"in"* | 0.7 | 1.7 | 0.3 | 0.3 | 2.4 | 1.7 | 2.0 |
| *"besides"* | 0.3 | 1.0 | 4.4 | 3.2 | 1.2 | 0.3 | 2.3 |
| *"does"* | 0.3 | 2.7 | 1.7 | 2.5 | 1.5 | 3.3 | 0.7 |
| other | 11.4 | 16.0 | 12.2 | 15.9 | 14.1 | 13.6 | 15.3 |

Table 4: **The distribution of question types** in MaXM across languages. Approximated by their corresponding English question prefixes.

erated by TransVQ²A[2]. Across languages, we observe at least 75% *Correct* or *Almost Correct* questions and, given these questions, at least 90% *Correct* or *Almost Correct* answers. This highlights the effectiveness of our approach.

**Answer Expansion and Standardization**. We split the generated questions into 4 categories: boolean, numeric, color, and others. We then asked the annotators to perform standardization of boolean, numeric, and color questions based on each language's guideline. For the rest of the questions, we tasked another set of at least 2 annotators per language to expand the answers to these questions with as many additionally correct (but not overly lengthy) answers as they can.

**Additional Filtering**. Our raters performed another round of verification, filtering out examples with "ambiguous" and "responsible-AI-sensitive" questions and/or with inappropriate image content. The raters also labeled "Collection" questions that are likely to lead to long answers that are difficult to evaluate, such as for "What is on the table?" when there are multiple items, without filter them out.

---

[2]For pairs generated by DirectQG, we did not perform exhaustive verification; we only asked the raters to annotate *Correct* and *Almost Correct* questions related to "no" answers.

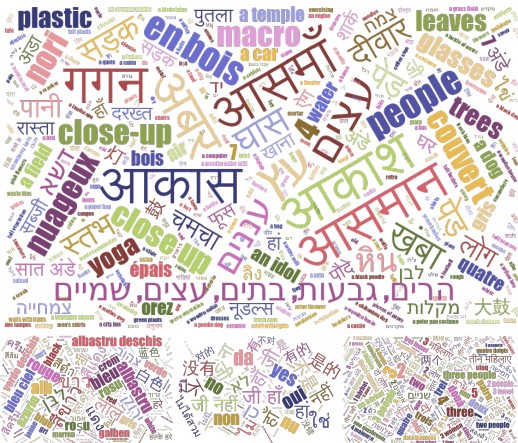

Figure 4: Top answer cloud is for *"What"* questions (excluding *"What color"*). Bottom answer clouds from left to right are for *"What color"*, Boolean *"Is/Are/Was/Were/Do/Does/Did"*, and *"How many"* questions, respectively.

## 4.4 Analysis and Discussion

**Size and Question Type and Answer Distributions**. MaXM v1 includes 2,142 questions in 7 languages: English (298), French (293), Hindi (294), Hebrew (315), Romanian (333), Thai (302), and Chinese (307).

Table 4 shows a breakdown of question types from MaXM. Since question prefixes in some languages are not indicative of question types (e.g., Thai does not always begin the "What" questions with the Thai "What"), we estimate a question's type using the prefix of its **English** version before translation. We observe diverse types and a high degree of linguistic variations. Fig. 4 presents word clouds for answers for selected question types: *"What"*, *"What color"*, Boolean, and *"How many"*, to further illustrate the diverse answers within MaXM for each question type.

**Comparison to xGQA**. In terms of settings, one difference is the languages of the answers; xGQA operates in the "cross-lingual" setting where the input question can be non-English but the output answer is always English. While this simplifies the evaluation process, we argue that the "multilingual" setting with non-English answers considered in

MaXM is more practical.

Another difference is the definition of the zero-shot setting; xGQA refers to unseen languages (not images) whereas our setting is more general, referring to both unseen images and languages. Finally, the type of translated data and how it is used for training are different; we only consider zero-shot setting and always use machine-translated questions for training, while xGQA considers both zero-shot and few-shot settings with human-translated questions involved only in the few-shot case.

In terms of the datasets, xGQA inherits the characteristics of GQA, whose questions are restricted in style (e.g., generated by a probabilistic template-based question engine) and in the skills required (e.g., reasoning-based with multi-step inference of object attributes and relationships) (Hudson and Manning, 2019). In contrast, MaXM's questions are more general. Additionally, xGQA considers the same set of questions for all languages, whereas MaXM considers different sets of questions guided by the captions in each language.

## 5 Evaluation

### 5.1 Evaluation Protocol

**Evaluation Metrics**. We use Exact Match *Accuracy* as the main evaluation measure for MaXM, following previous work on VQA (Antol et al., 2015; Goyal et al., 2017; Gurari et al., 2018). We deem the answer as correct if it matches any of the ground-truth answers. To assess the degree of strictness of this measure, we also consider *soft* text similarity metrics *CIDEr* (Vedantam et al., 2015) and *ROUGE-L* (Lin, 2004) in our experiments, where we treat each of the ground-truth answers equally as one of the references (as if each of them was answered by an annotator).

**Training Data**. MaXM is a test-only benchmark; it cannot be used for training. We designate VQA2.0 (Goyal et al., 2017) and its translations as the default training data source for our benchmark, due to its popularity and quality, similarly to the use of COCO-Captions (Chen et al., 2015) for the nocaps benchmark (Agrawal et al., 2019) in the image captioning task. Nevertheless, we allow free use of existing VQA resources for training as long as the corresponding training images do not overlap with MaXM images. In our experiments, we also consider VQ$^2$A-COCO and VQ$^2$A-CC3M (Chang-pinyo et al., 2022) to assess the effect of text domain gap.

### 5.2 Models for Multilingual VQA

Inspired by approaches to multilingual NLP research, we consider two main families of models for mVQA that adapt existing source English VQA datasets to target languages: *Translate-Test* and *Translate-Train*. Translate-Test leaves the training data and the model as-is, but translates the test VQA data to the the source language English, apply the model, and then translate it back to the target language. On the other hand, *Translate-Train* translates the English VQA data to a target language, trains a model on this pseudo VQA data (i.e., their translations), and directly apply the trained model to the test data.

**Translate-Test**. We consider two open-source state-of-the-art VQA models: **OFA-Large** (Wang et al., 2022b) and BLIP2 (Li et al., 2023). Neither of them are designed for mVQA.

**Translate-Train**. We include the results from the state-of-the-art multilingual vision-and-language model **PaLI-17B** (Chen et al., 2023), which pretrains on diverse VQA datasets in 35 languages (Thapliyal et al., 2022) among other datasets, and then finetune on VQA2.0 in 13 languages: en, bn, de, fr, hi, id, iw, ko, pt, ro, ru, th, zh. Further, we implement a lightweight version of PaLI, called Simple Multi-Language Prompted Training, **Simple MPT**, with a much smaller model and without vision-and-language pre-training. **Simple MPT** is trained on the data in 13 languages in a multi-task fashion. Details can be found in Appendix C.

### 5.3 Results

**Main Results**. Table 5 benchmarks our proposed Simple MPT and state-of-the-art VQA models on MaXM. We observe that PaLI-17B performs best on all languages. This can be attributed to both the fact that PaLI is the strongest English VQA model and the fact that it was designed to be multilingual, leveraging pre-training image-text corpus in 105 languages. This result suggests it can be beneficial to design and develop multilingual VQA models from day one.

Surprisingly, our proposed Simple MPT model is a strong baseline even though it is much smaller than PaLI and does not leverage multilingual pre-training data. While its English performance is on par with OFA and much worse than BLIP2, its multilingual performance excels, outperforming

| Model | | | Language | | | | | | |
|---|---|---|---|---|---|---|---|---|---|
| | | | en | fr | hi | iw | ro | th | zh |
| *Translate-Test* | OFA (Wang et al., 2022b) | [470M] | 35.6 | 13.3 | 41.5 | 31.7 | 26.4 | 29.5 | 17.6 |
| | BLIP2 (Li et al., 2023) | [11B] | *48.7* | 20.1 | *63.3* | *49.2* | 39.0 | 49.0 | 28.0 |
| *Translate-Train* | Simple MPT (ours) | [1.5B] | 36.6 | *36.2* | 55.1 | 40.6 | *42.3* | *50.0* | *30.3* |
| | PaLI (Chen et al., 2023) | [17B] | **56.4** | **46.4** | **67.3** | **60.0** | **57.4** | **65.6** | **46.9** |

Table 5: **Benchmarking VQA models on** MaXM. Accuracy (%) of OFA-Large (OFA), BLIP2, our lightweight Simple MPT, and PaLI-17B (PaLI), with approximate parameter count in brackets. All are finetuned on VQA2.0, English-only for *Translate-Train* and 13 languages for Translate-Test. Best results are **bold**. Second best *italized*.

| Model | | Training Dataset | Language | | | | | | |
|---|---|---|---|---|---|---|---|---|---|
| | | | en | fr | hi | iw | ro | th | zh |
| *Translate-Train* | Single-Language | VQA2.0 | 37.6 | 33.8 | 53.7 | 35.6 | 36.0 | 50.0 | 29.0 |
| | Simple MPT | VQA2.0 | 36.6 | 36.2 | 55.1 | 40.6 | 42.3 | 50.0 | 30.3 |
| | Simple MPT | VQ$^2$A-COCO | **48.0** | **43.3** | **56.8** | **42.2** | **45.6** | **52.3** | **34.5** |
| | Simple MPT | VQ$^2$A-CC3M | 38.3 | 34.1 | 45.6 | 34.9 | 36.9 | 45.0 | 29.0 |

Table 6: **Effect of Training Data Sources**. Accuracy of Single-Language baselines (MPT architecture) and Accuracy (%) of MPT models trained on different training datasets.

OFA in all languages and underperforms BLIP2 only for Hindi and Hebrew.

Overall, our result suggests that *Translate-Train* may be a superior approach to mVQA to *Translate-Test*. We note, however, that in our early experiments, we find that Translate-Train is inferior to Translate-Test as an adaptation approach for *English* VQA models. For instance, the answer of finetuned BLIP2 to the French question *"Outre les fleurs roses, quelle autre couleur y avait-il dans le jardin?"* (*"Besides pink flowers, what other color was there in the garden?"*) is *"pink"* while the correct answer is *"blanc"* (*"white"*) — wrong both in terms of language and semantics. It is not immediately obvious how to adapt English VQA models with, for example, vocab and tokenizers that overfit the English language. This again suggests that the design of these multimodal models would benefit from having multilinguality in mind from the start.

**Single-Language vs. Multi-Language Training, Different Training Datasets**. In Table 6, our Simple MPT model performs similarly or better than each of the Single-Language baselines. This suggests that modern models are capable of learning from related languages. We also find that translated COCO is overall the best training data source. We attribute this to (i) the fact that VQ$^2$A was used to generate VQ$^2$A-COCO, and (ii) VQ$^2$A-COCO is generally more robust in the cross-dataset setting (Changpinyo et al., 2022). However, VQ$^2$A-CC3M is unable to outperform VQA2.0 despite (i); applying VQ$^2$A to the noisy alt-texts in CC3M (Sharma et al., 2018) is prone to errors that would only be exacerbated by automatic MT.

**Less Strict Metrics**. In Table 7 We observe generally consistent results when using CIDEr and ROUGE-L instead of the stricter Accuracy, except for Thai and Chinese, where the gaps in Accuracy are small to begin with.

**No Adaptation via Translate-Test**. Can existing English VQA models *work* out of the box? In Table 8, we find that the answer is no. Expectedly, the models perform well on French, which is closer to English than other languages are.

**Simple MPT on xGQA**. Can our modeling approach be extended to the cross-lingual setting in xGQA (Pfeiffer et al., 2022)? We report this result in Appendix D.

# 6 Conclusions

We take initial steps toward multilingual VQA by proposing scalable solutions on both data creation and modeling fronts. We create a multilingual VQA benchmark in 7 diverse languages to drive modeling progress on multilingual VQA. We establish strong unified and open-ended VQA models that work well on 13 languages as well as benchmark state-of-the-art models. For future work, we would like to expand native-language question generation that is done in a limited scope and have single one for all target answers.

**Acknowledgments**. We would like to thank Jialin Wu, Kenton Lee, Tomer Levinboim, Nan Ding, Sarah Laszlo, Doron Kukliansky, and Tania Bedrax-Weiss for their feedback and discussion. Word clouds are generated from https://www.jasondavies.com/wordcloud/.

| Metric | Model | | Language | | | | | | |
|---|---|---|---|---|---|---|---|---|---|
| | | | en | fr | hi | iw | ro | th | zh |
| Accuracy | *Translate-Train* | Simple MPT | 36.6 | **36.2** | 55.1 | 40.6 | **42.3** | **50.0** | **30.3** |
| | *Translate-Test* | OFA | 35.6 | 13.3 | 41.5 | 31.7 | 26.4 | 29.5 | 17.6 |
| | | BLIP2 | **48.7** | 20.1 | **63.3** | **49.2** | 39.0 | 49.0 | 28.0 |
| CIDEr | *Translate-Train* | Simple MPT | 91.5 | **102.0** | 62.0 | 78.6 | **89.6** | 86.7 | **68.4** |
| | *Translate-Test* | OFA | 88.3 | 49.4 | 66.3 | 78.6 | 60.4 | 70.2 | 51.1 |
| | | BLIP2 | **121.8** | 67.6 | **88.1** | **93.8** | 79.1 | **91.5** | 65.7 |
| ROUGE-L | *Translate-Train* | Simple MPT | 45.0 | **47.9** | 57.9 | 42.8 | **49.4** | **57.7** | 37.7 |
| | *Translate-Test* | OFA | 47.5 | 27.2 | 52.0 | 44.8 | 38.3 | 44.0 | 31.4 |
| | | BLIP2 | **62.6** | 32.6 | **67.2** | **52.6** | 47.4 | 53.6 | **39.9** |

Table 7: **Results on Soft Metrics**. Accuracy (%), CIDEr ($\times$ 100), and ROUGE-L ($\times$ 100) of Simple MPT, OFA, and BLIP2. All are finetuned on VQA2.0. Best results are **bold**.

| Metric | Model | | Language | | | | | | |
|---|---|---|---|---|---|---|---|---|---|
| | | | en | fr | hi | iw | ro | th | zh |
| Accuracy | *Translate-Test* | OFA | 35.6 | 13.3 | 41.5 | 31.7 | 26.4 | 29.5 | 17.6 |
| | | BLIP2 | 48.7 | 20.1 | 63.3 | 49.2 | 39.0 | 49.0 | 28.0 |
| | No adapt | OFA | 35.6 | 0.3 | 0.3 | 0.3 | 4.8 | 0.0 | 0.0 |
| | | BLIP2 | 48.7 | 9.2 | 0.3 | 1.9 | 3.9 | 7.0 | 2.3 |
| CIDEr | *Translate-Test* | OFA | 88.3 | 49.4 | 66.3 | 78.6 | 60.4 | 70.2 | 51.1 |
| | | BLIP2 | 121.8 | 67.6 | 88.1 | 93.8 | 79.1 | 91.5 | 65.7 |
| | No adapt | OFA | 88.3 | 4.8 | 3.7 | 4.8 | 12.6 | 7.8 | 3.6 |
| | | BLIP2 | 121.8 | 28.0 | 5.1 | 4.0 | 12.1 | 16.0 | 8.2 |
| ROUGE-L | *Translate-Test* | OFA | 47.5 | 27.2 | 52.0 | 44.8 | 38.3 | 44.0 | 31.4 |
| | | BLIP2 | 62.6 | 32.6 | 67.2 | 52.6 | 47.4 | 53.6 | 39.9 |
| | No adapt | OFA | 47.5 | 2.6 | 4.4 | 4.8 | 5.9 | 7.3 | 6.6 |
| | | BLIP2 | 62.6 | 12.5 | 6.2 | 2.9 | 5.4 | 10.8 | 7.3 |

Table 8: **Results without adaptation for Translate-Test**. Accuracy (%), CIDEr ($\times$ 100), and ROUGE-L ($\times$ 100) of OFA and BLIP2 with Translate-Test and without (No adapt).

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

## General

- For each row, there will be an image, a question, and an answer for you to rate.
- An image may have multiple questions and an image-question pair may have multiple answers; in both cases, we split them into multiple rows.
- Note on locale/region: Use the most common locale/region.
- Note on i18n keyboard with respect to diacritics:
  - Do use the strict/correct i18n characters (with correct diacritics, etc.) when providing a correct question/answer;
  - Do **not** penalize the question and/or answer because of missing/incorrect diacritics when assessing their correctness.

## Question Annotation

| | |
|---|---|
| **Correct** | The question is **Correct** if all of the following are true:
i) It **makes sense**.
ii) It is **relevant** to the image.
iii) It is NOT **Ambiguous** or **RAI sensitive**. (See the definitions below).

**"Relevant"**
• Must require an image to answer (e.g., not "Is dimsum Chinese?")
• Does not require too specialized knowledge (e.g., not "How fast can this type of plane go?")
• Does not require the knowledge of the photo taker (e.g, not "Is it my son's birthday?")
• Does not contain ambiguous references (e.g., not "What are they doing?" when it is unclear who "they" refers to)
• Note: As long as you can answer the question based on the image, the question is relevant, for example, even when the question asks about the existence or the cardinality of an object X but the object X is not present in the image. |
| **Almost Correct** | The question is **Correct** but its surface form can be improved — there are syntactic errors or awkward/uncommon usages. This includes
• Wrong word choice
• Wrong word ordering
• [Lang specific] Wrong or missing function words
• [Lang specific] Wrong conjugation or morphological form
• [Lang specific] Missing affixes ((a) missing prepositional prefix "לצלחת vs בצלחת"; (b) missing determiner prefix "רוטב vs הרוטב" and "מארת מחשב" vs. "מארת המחשב")
• [Lang specific] Inappropriate tense. Stick to the present tense unless the tense matches the caption.
• [Lang specific] Not neutral
Feel free to also use this to remove RAI sensitive terms, such as those related race (e.g., an African kid) |
| **Ambiguous** | The question is highly subjective, likely leading to **answers that are unrelated or are in high disagreement** ("Is the room tidy?" when it is unclear if it is one way or the other) |
| **Incorrect** | The question does not fall into any of the above. Usually, this means the question does NOT make sense OR is NOT relevant to the image. |
| **RAI** | The question is RAI sensitive. See What counts as RAI-sensitive? |

When the question is Correct or Almost Correct, see if the question falls into "Collection":

| | |
|---|---|
| **Collection** | The question will likely create **a large number of possible answers that are categorically different** ("What is on the table?" when there are more than 3 things on the table) |

### What counts as RAI-sensitive?

- Be aware of **protected categories** included race, color, ethnic or national origin; age; religion or religious creed (or belief, where applicable); sex, including pregnancy, childbirth, breastfeeding, or related medical conditions; sexual orientation; gender, gender identity, gender expression, transgender status, or sexual stereotypes; nationality, immigration status, citizenship, or ancestry; marital status; protected military or veteran status; physical or mental disability, medical condition, genetic information or characteristics (or those of a family member); status as a victim of domestic violence, sexual assault or stalking; or any other basis prohibited under federal, state, or local law.
- The question and the answer should NOT contain a person's race ("African", "Asian"). Flag as RAI or remove it using Almost Correct. Note that "a dark-skinned person" or "a black person" are OK.
- If the question **assumes** a protected group (gender/age etc.) and you can validate that it is correct, then it is OK; but, if the question assumes a protected group and it is wrong or unclear, please mark RAI or fix it using Almost Correct..
- If the question asks to **predict** or **describe a trait** of a protected group, mark RAI no matter what the answer is.
- "Who" question
  - Similar to above, it is NOT OK to use race to refer to people ("African", "Asian")
  - Similar to above, it is OK to use gender or age to refer people since the question's intent is not for predicting or describing a trait of a protected group

Figure 5: **Detailed Instructions on Question Annotation**

# A  Considerations and Limitations

Our dataset is intended to be used for research-only purposes.

Our pipeline takes in an image caption as input. Image captions may have mistakes and biases, which could be further amplified by machine learning models used by our approach. In particular, we use generative models for automatic question generation and machine translation that may create outputs with incorrect or nonfactual contents or outputs with Translationese artifacts. We have mitigated this manually via human in the loop and automatically via the caption-answer consistency check (cf., Sect. 3.2). Note that the English VQ²A (Changpinyo et al., 2022) that we leverage in our pipeline also has similar filtering using the round-trip consistency check via question answering. Together these significantly improve the cor-

## Answer Annotation

- Unless the question is **Ambiguous**, **Incorrect**, or **RAI**, **ALWAYS** provide the correct answer, even when the answer is **Incorrect** or **Almost Correct**.
- If you are unsure if the answer is correct, please select **Incorrect** and provide an explanation. Note that if the answer requires too specialized knowledge, you should have marked the *question* as **Incorrect**.

| | |
|---|---|
| **Correct** | The answer is **Correct** if it satisfies **the question's intent**. That is, the answer **must be specific enough** (e.g., "What is this?" "Soda" or "Coca Cola" is OK but "An item" is not.) |
| **Almost Correct** | The answer is **Correct** but its surface form can be improved. This includes
• Wrong word choice
• Missing preposition ("table" vs. "on the table")
• [Lang specific] Missing articles
• [Lang specific] Wrong morphological form (masculine vs. feminine)
• Answer type mismatch (Q: "Which color is the plate?" A: "white plate" instead of "white" to match "which color")
• Generation noise (Q: "How many wheels?" A: "three wheels g")
• Missing quantifier ("clouds" vs. "a few clouds")
• Partial answer (Q: "Where are the stones?" A: "in front of the woman" instead of "in front of the woman and behind her")
• Missing noun/verb head  (Q: "Which piles are in the image" A: "of chairs"  vs. "piles of chairs" or "chairs")
• Too long
• Contains unnecessary punctuations |
| **Incorrect** | The answer does not fall into **Correct** or **Almost Correct**. |
| **N/A** | Used when the question is **Ambiguous**, **Incorrect**, or **RAI**. |

### Answer Expansion and/or Standardization

Provide other possible answers. Separate each answer by " ||| ".

There can be multiple surface forms that are correct. Thus, we provide a guideline for major question types below.
- **"Yes/No": Be as brief as possible.**
  - English: "yes" or "no" (not "yes, it is")
  - Thai: In some cases, additional answers are more natural.
    For "Is there/Are there" questions → people also say "มี" ("There is/are")
    "ไม่" and "ไม่ใช่" are both "no."
- **"Number": Use both digit and text with and without a short noun. Do not add adjectives.**
  - English: "2", "two", "2 people", "two people" (use a short noun, not "two tall people")
  - Thai: Additional pronouns (depending on the nouns) as an additional suffix to "2" and "two"
    Example: "คน" are used for all people, "ดอก" are used for all flowers
- **"Color": Use a short but syntactically correct form.**
  - English: "green" (not "green trees"), "black and white" and "white and black"
  - Thai: Also add the word "color" as a prefix.
    We say "สีเขียว" ("color green") or "เขียว" ("green").
- **"Where": Absolute or relative positions are both OK.**
  Where is the table? "next to a sofa" and "in a room" are both valid.
  Where is the car? "next to the tree" and "on a road" are both valid.
- **"What kind": Use your best judgment; some are "Ambiguous" and some are OK.**

Figure 6: **Detailed Instructions on Answer Annotation as well as Answer Expansion and Standardization.**

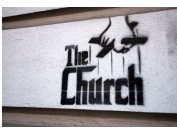

Q: Y a-t-il une croix sur le mur?
A: oui
*(Q: is there a cross on the wall?*
*A: yes)*

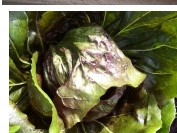

Q: तस्वीर में कौन से साग हैं?
A: सलाद पत्ता
*(Q: What greens are in the picture?*
*A: lettuce)*

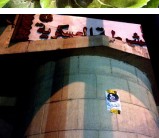

Q: האם יש מבנה עם שלטים בערבית?
A: כן
*(Q: Is there a building with signs in Arabic?*
*A: yes)*

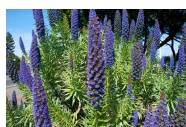

Q: Ce fel de flori înfloresc pe marginea drumului?
A: lupin, flori de lupin albastru, flori mov, flori mov şi lunguieţe, flori lunguieţe, mari şi mov
*(Q: What kind of flowers bloom on the side of the road?*
*A: lupine, blue lupine flowers, purple flowers, long purple flowers, long large purple flowers)*

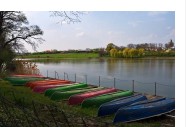

Q: เรืออยู่ที่ไหน?
A: ริมทะเลสาบ, ริมแม่น้ำ, ริมฝั่ง, ริมตลิ่ง, บนบก, บนฝั่ง, ข้างทะเลสาบ, ข้างแม่น้ำ, ข้างฝั่ง, ข้างตลิ่ง
*(Q: Where are the boats?*
*A: by the lake, by the river, shore, along the bank, on land, by the lake, by the river, shore, beside the bank)*

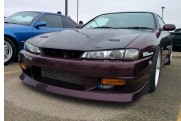

Q: 紫色车左边的车是什么颜色的?
A: 蓝色, 蓝色的
*(Q: What color is the car to the left of the purple one?*
*A: blue, blue)*

Figure 7: **Additional MaXM examples.**

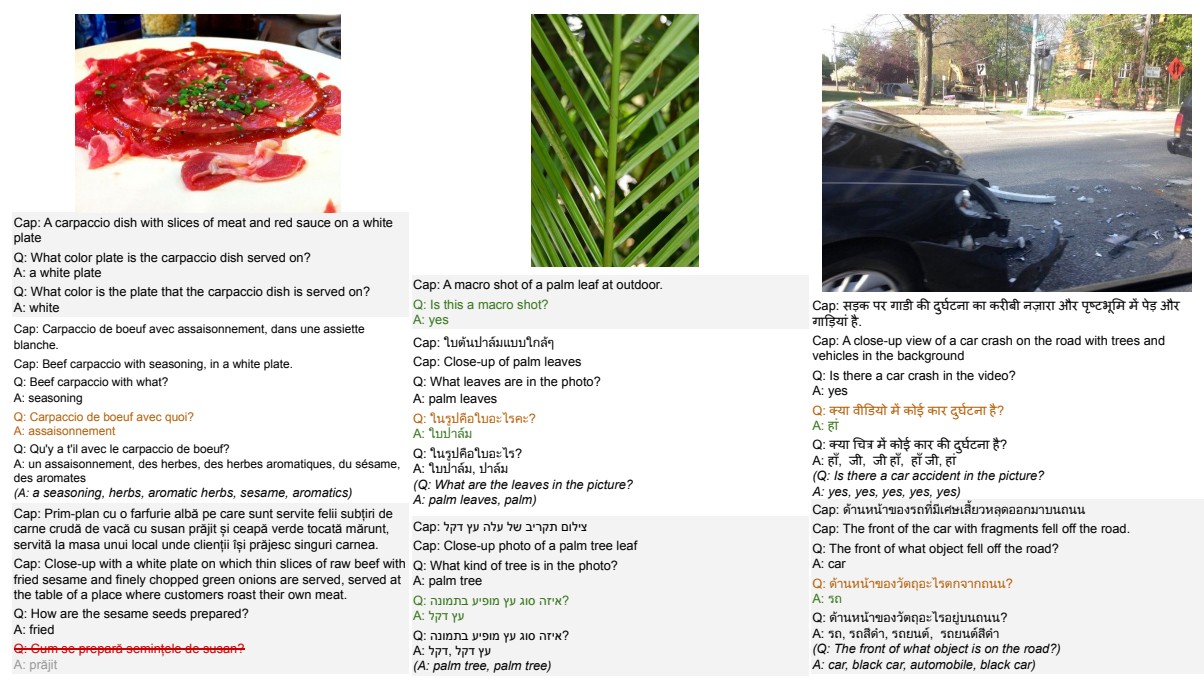

Cap: A carpaccio dish with slices of meat and red sauce on a white plate
Q: What color plate is the carpaccio dish served on?
A: a white plate
Q: What color is the plate that the carpaccio dish is served on?
A: white

Cap: Carpaccio de boeuf avec assaisonnement, dans une assiette blanche.
Cap: Beef carpaccio with seasoning, in a white plate.
Q: Beef carpaccio with what?
A: seasoning
Q: Carpaccio de boeuf avec quoi?
A: assaisonnement
Q: Qu'y a t'il avec le carpaccio de boeuf?
A: un assaisonnement, des herbes, des herbes aromatiques, du sésame, des aromates
(A: a seasoning, herbs, aromatic herbs, sesame, aromatics)

Cap: Prim-plan cu o farfurie albă pe care sunt servite felii subțiri de carne crudă de vacă cu susan prăjit și ceapă verde tocată mărunt, servită la masa unui local unde clienții își prăjesc singuri carnea.
Cap: Close-up with a white plate on which thin slices of raw beef with fried sesame and finely chopped green onions are served, served at the table of a place where customers roast their own meat.
Q: How are the sesame seeds prepared?
A: fried
Q: Cum se prepară semințele de susan?
A: prăjit

Cap: A macro shot of a palm leaf at outdoor.
Q: Is this a macro shot?
A: yes

Cap: ใบต้นปาล์มแบบใกล้ๆ
Cap: Close-up of palm leaves
Q: What leaves are in the photo?
A: palm leaves
Q: ในรูปคือใบอะไรคะ?
A: ใบปาล์ม
Q: ในรูปคือใบอะไร?
A: ใบปาล์ม, ปาล์ม
(Q: What are the leaves in the picture?
A: palm leaves, palm)

Cap: צילום תקריב של עלה עץ דקל
Cap: Close-up photo of a palm tree leaf
Q: What kind of tree is in the photo?
A: palm tree
Q: איזה סוג עץ מופיע בתמונה?
A: עץ דקל
Q: איזה סוג עץ מופיע בתמונה?
A: עץ דקל, דקל
(A: palm tree, palm tree)

Cap: सड़क पर गाड़ी की दुर्घटना का करीबी नज़ारा और पृष्ठभूमि में पेड़ और गाड़ियां है.
Cap: A close-up view of a car crash on the road with trees and vehicles in the background
Q: Is there a car crash in the video?
A: yes
Q: क्या वीडियो में कोई कार दुर्घटना है?
A: हाँ
Q: क्या चित्र में कोई कार की दुर्घटना है?
A: हाँ, जी, जी हाँ, हाँ जी, हाँ
(Q: Is there a car accident in the picture?
A: yes, yes, yes, yes, yes)

Cap: ด้านหน้าของรถที่มีเศษเสี้ยวหลุดออกมาบนถนน
Cap: The front of the car with fragments fell off the road.
Q: The front of what object fell off the road?
A: car
Q: ด้านหน้าของวัตถุอะไรตกจากถนน?
A: รถ
Q: ด้านหน้าของวัตถุอะไรอยู่บนถนน?
A: รถ, รถสีดำ, รถยนต์, รถยนต์สีดำ
(Q: The front of what object is on the road?)
A: car, black car, automobile, black car)

Figure 8: **Additional examples on our approach to multilingual VQA data generation**. Green, yellow, and red texts correspond to "Correct", "Almost Correct", and "Incorrect," respectively.

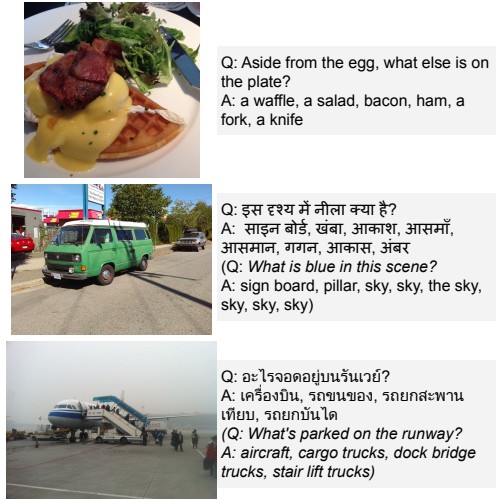

Q: Aside from the egg, what else is on the plate?
A: a waffle, a salad, bacon, ham, a fork, a knife

Q: इस दृश्य में नीला क्या है?
A: साइन बोर्ड, खंबा, आकाश, आसमाँ, आसमान, गगन, आकास, अंबर
(Q: What is blue in this scene?
A: sign board, pillar, sky, sky, the sky, sky, sky, sky)

Q: อะไรจอดอยู่บนรันเวย์?
A: เครื่องบิน, รถขนของ, รถยกสะพานเทียบ, รถยกบันได
(Q: What's parked on the runway?
A: aircraft, cargo trucks, dock bridge trucks, stair lift trucks)

Figure 9: **Examples of Collection questions.**

rectness and fluency of our pipeline. In addition, we explicitly mark examples that can be considered Responsible-AI-sensitive, but not necessarily incorrect; see Sect. B for details and examples.

Another type of biases is the low coverage of particular types of answers, resulting from the image captions not mentioning the absences of objects or properties. We have also taken a step toward mitigating this. See Sect. 3.3.

Finally, we select a diverse set of languages, alleviating typological, genealogical, and geographical language biases presented in the VQA research community.

We mainly use Crossmodal-3600

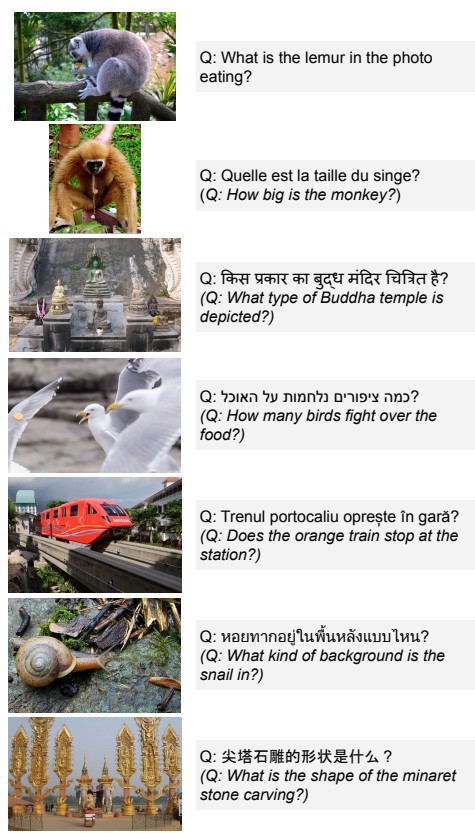

Q: What is the lemur in the photo eating?

Q: Quelle est la taille du singe?
(Q: How big is the monkey?)

Q: किस प्रकार का बुद्ध मंदिर चित्रित है?
(Q: What type of Buddha temple is depicted?)

Q: כמה ציפורים נלחמות על האוכל?
(Q: How many birds fight over the food?)

Q: Trenul portocaliu oprește în gară?
(Q: Does the orange train stop at the station?)

Q: หอยทากอยู่ในพื้นหลังแบบไหน?
(Q: What kind of background is the snail in?)

Q: 尖塔石雕的形状是什么？
(Q: What is the shape of the minaret stone carving?)

Figure 10: **Examples of Ambiguous questions** that we flagged and filtered out.

(XM3600) (Thapliyal et al., 2022). Open Images (Krasin et al., 2017; Kuznetsova et al., 2020) and the multilingual captions in XM3600 are human-curated and cleaned, which mitigates the risks that MaXM would contain information

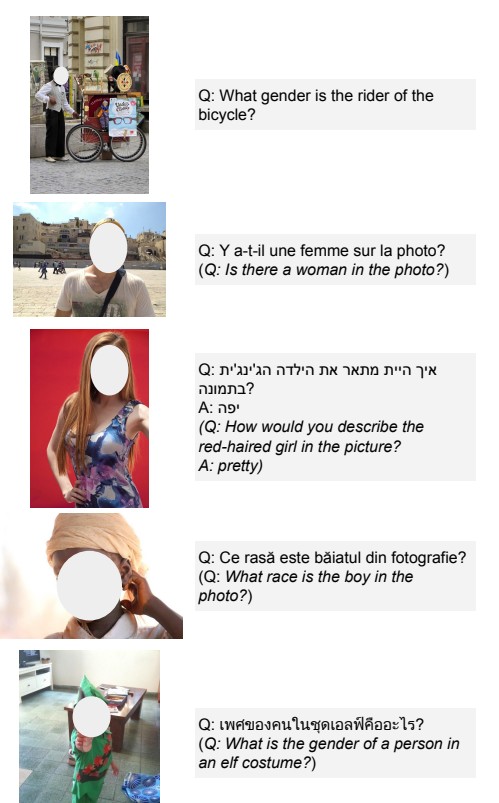

Q: What gender is the rider of the bicycle?

Q: Y a-t-il une femme sur la photo?
(*Q: Is there a woman in the photo?*)

Q: איך היית מתאר את הילדה הג׳ינג׳ית בתמונה?
A: יפה
(*Q: How would you describe the red-haired girl in the picture?*
*A: pretty*)

Q: Ce rasă este băiatul din fotografie?
(*Q: What race is the boy in the photo?*)

Q: เพศของคนในชุดเอลฟ์คืออะไร?
(*Q: What is the gender of a person in an elf costume?*)

Figure 11: **Examples of Responsible-AI-sensitive questions** that we flagged and filtered out. Faces are hidden.

that names or uniquely identifies individual people or offensive content.

## B  Human Verification and Modification

### B.1  Annotation Guideline

We provide our general instructions and detailed instructions on question annotation in Fig. 5, where we explicitly ask the annotators to be wary of Responsible-AI-sensitive questions. Fig. 6 provides detailed instructions on answer annotation and on answer expansion and standardization.

### B.2  Additional Examples

**Additional Examples**. Fig. 7 provides additional examples to the ones in Fig. 1. Again, we highlight the richness and diversity of our questions. For instance, it requires recognizing a cross under occlusion (French), a type of vegetables (Hindi), the Arabic language (Hebrew), and a type of flowers (Romanian). Some of these examples are specific to particular languages; it would be difficult for other language speakers to answer the Hebrew example (or the Chinese example in Fig. 1, which requires OCR).

We also highlight the richness of our candidate answers. For the "where" question in Thai, 10 answers count as correct. Similarly, the Romanian example in Fig. 1 provides multiple diverse surface forms for "coffee with cream."

Fig. 8 additional examples to the Chinese one in Fig 2. These examples showcase the efficiency of our annotation process. They also provide concrete examples of "Almost Correct." For instance, in the middle example, the Thai translation of "What leaves are in the photo?" is not *neutral* because it contains an Honorific particle [3]; it ends with "khá" which signifies a sign of respect to the addressee and indicates that the sex of the speaker is female. Finally, these examples provide a glimpse of sources of errors. For instance, it is VQ$^2$A that hallucinates "in the video" in the Hindi example on the right.

**Collection Examples**. Fig. 10 provides examples of "Collection" questions. We keep these questions as we believe they are useful in practice and as a way to encourage the community to work on better automatic evaluation metrics for this type of questions.

**Ambiguous Examples**. Fig. 9 provides examples of "Ambiguous" questions that we filter out. Reasons include object being too small (English) or irregular (Chinese), determining sizes being subjective (French), and not enough context (Hebrew, Romanian). "What kind/What type" questions are particularly difficult to answer and tend to be ambiguous.

**Responsible-AI-sensitive Examples**. Fig. 11 provides examples of Responsible-AI-sensitive questions that we filter out. These cases are often associated with directly asking for the information about or describing a particular gender or race, or involving an incorrect assumption about such protected attributes (e.g., girl vs. woman in the Hebrew example).

## C  Simple MPT

In this section, we describe **Simple MPT**, a lightweight model for mVQA in detail.

**Design**. Much of the previous work on VQA is built for English. Further, VQA is often formulated as *vocab-based VQA*, a classification task into a pre-defined space of top (English) answer vocabulary; see, e.g., (Antol et al., 2015; Goyal et al.,

---

[3] https://en.wikipedia.org/wiki/Thai_honorifics

| Model | Finetuning Dataset | Question Language | | | | | | | |
|---|---|---|---|---|---|---|---|---|---|
| | | en | bn | de | id | ko | pt | ru | zh |
| M3P (Pfeiffer et al., 2022) | GQA | **58.4** | 17.6 | 24.8 | 18.7 | 19.7 | 26.7 | 24.3 | 19.7 |
| mBERT[Ada] (Pfeiffer et al., 2022) | GQA | *56.3* | 13.4 | 32.4 | 19.8 | 19.9 | 31.5 | 25.5 | 26.2 |
| Single-Language | VQA2.0 | 43.1 | 37.9 | 39.6 | **40.4** | *38.9* | *40.3* | 39.3 | *39.7* |
| Simple MPT | VQA2.0 | 41.5 | *38.6* | *40.5* | 39.5 | 38.7 | 39.8 | *39.5* | 39.5 |
| Simple MPT | VQ²A-COCO | 36.6 | 34.3 | 36.1 | 35.5 | 35.1 | 34.6 | 34.5 | 35.4 |
| Simple MPT | VQ²A-CC3M | 34.0 | 30.9 | 33.3 | 33.2 | 32.5 | 32.1 | 32.0 | 32.7 |
| PaLI (Chen et al., 2023) | VQA2.0 | 54.2 | **50.0** | **52.2** | **50.6** | **50.4** | **51.3** | **50.3** | **50.6** |

Table 9: **Zero-Shot Results on xGQA**. Accuracy (%) of our Simple MPT models trained on different training datasets as well as our Single-Language baselines and the baselines from (Pfeiffer et al., 2022). Best results are **bold**. Second best *italicized*.

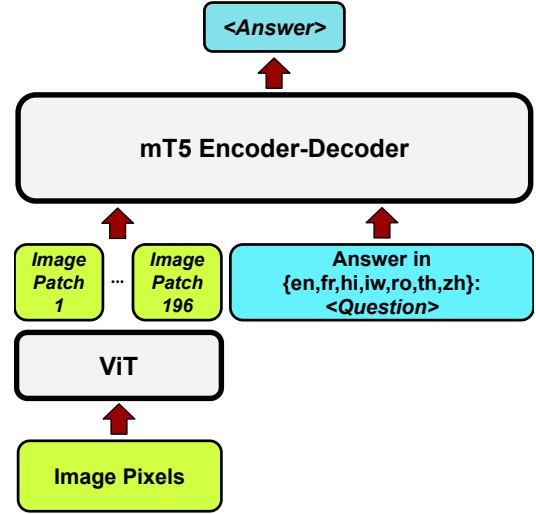

Figure 12: **Our Simple MPT model** used in our experiments. We leverage ViT (Dosovitskiy et al., 2021) and mT5 (Xue et al., 2021) and train them together end-to-end.

2017). The main drawback of this approach is its inability to deal with rare answers through language compositionality. Recent work considers VQA as generation (Cho et al., 2021; Wang et al., 2022c; Alayrac et al., 2022; Wang et al., 2022a), capable of *open-ended VQA*. We adopt this as a scalable and flexible modeling approach to mVQA as the language coverage increases. In particular, we propose a *single* open-ended VQA model for multiple languages. Our proposed formulation is more desirable than existing ones since it takes advantage of both compositionality in individual languages and the relationship among related languages. To this end, we first describe an encoder-decoder architecture for VQA in the open-ended generation setting. Then, we describe how we train this model for multiple languages. This is summarized in Fig. 12.

**Open-Ended VQA**. Our starting architecture is mT5 (Xue et al., 2021), a multilingual variant of T5 (Raffel et al., 2020). mT5 is an encoder-decoder transformer-based architecture, pre-trained on a Common Crawl-based dataset covering 101 languages. This allows us to leverage multilingual language understanding (for the questions) and generation (for the answers) from the get-go. To adapt mT5 to the VQA task, we prepend patch embeddings from the image to the question tokens. In particular, we encode the image pixels using Vision Transformers (ViT) (Dosovitskiy et al., 2021). We use ViT-L16 and mT5-Large in all of our experiments. Both mT5 and ViT are trained together in an end-to-end fashion to predict the target answer for each image-question pair, using the standard cross-entropy loss.

**Multi-Language Prompted Training**. We resort to multi-task prompted/instruction training (Sanh et al., 2022; Wei et al., 2022), where a task corresponds to VQA for a particular language. For the input question ⟨question⟩ in language ⟨lang⟩, we construct the prompt "Answer in ⟨lang⟩: ⟨question⟩" and use it as the text input to our model, similar to a modification to the input in Google's Multilingual Neural Machine Translation System (Johnson et al., 2017b). Such a design for multi-task learning makes extending VQA to multiple languages simple; as data for additional languages become available, one can simply add them to the pool without the need for architecture changes.

**Implementation Details**. We use the Flax implementation (Bradbury et al., 2018). For training both our 2⟨lang⟩ and 2en models, we use Adafactor (Shazeer and Stern, 2018) with a $\beta_1$ of 0 and a second-moment exponential decay of 0.8. We use a linear warmup of 1K steps with a peak learning of learning rate of 1e-3 and inverse square-root decay. We set the ViT dropout rate to 0 and the mT5 dropout rate to 0.1. We train each model with data parallelism using 16 Cloud TPU Pods[4], each with a batch size of 512, for 100K steps. We use

---

[4]https://cloud.google.com/tpu

standard image resolution of 224x224. We use the maximum input length of 24 and the target output length of 8.

We consider three datasets for training Simple MPT, all are translations of existing large-scale English VQA datasets to the 13 languages covered by MaXM and xGQA. We use the Karpathy training split (Karpathy and Fei-Fei, 2015) for VQA2.0 and VQ$^2$A-COCO and the standard training split for VQ$^2$A-CC3M.

## D   Additional Results

Our Simple MPT in the main paper predicts the answer in the same language as the question. Here, we explore if our Simple MPT can also be useful for the cross-lingual setting in xGQA (Pfeiffer et al., 2022), where the model always predicts the answer in English.

Similar to MaXM, xGQA is a test-only benchmark, the testdev split of 12,578 question-answer pairs per language from 398 images in 8 languages (en,bn,de,id,ko,pt,ru,zh). To evaluate Simple MPT on this dataset, we use the setting in the main paper: training on VQA2.0, VQ$^2$A-COCO, and VQ$^2$A-COCO but do not translate the training answers. We also use the prompt "Answer in en: ⟨question⟩" instead of "Answer in ⟨lang⟩: ⟨question⟩".

Table 9 reports the results. Our baselines are M3P and mBERT$^{Ada}$ from (Pfeiffer et al., 2022). Note that both M3P and mBERT$^{Ada}$ have access to the (English) GQA training data (Hudson and Manning, 2019), where our model does not. On the other hand, they do not use translated data as in our case. We outperform *multilingual* zero-shot baselines on all non-English languages, without access to English GQA labeled data. This further confirms that our unified approach to mVQA is effective. In addition, unlike on MaXM, VQA2.0 is the best pre-training data source. We attribute this to the fact that VQA2.0 and xGQA share COCO images (Lin et al., 2014). This highlights the utility of MaXM as additional out-of-domain test-only VQA evaluation data.