# OpenReview forum: "MaXM: Towards Multilingual Visual Question Answering"
_EMNLP/2023/Conference — EMNLP 2023 Findings_

### Official Review · Reviewer_jdG6 · 2023-08-01

**Paper Topic And Main Contributions:** 1. This work contributes a novel mult…
**Soundness:** 3

**Excitement:**

3: Ambivalent: It has merits (e.g., it reports state-of-the-art results, the idea is nice), but there are key weaknesses (e.g., it describes incremental work), and it can significantly benefit from another round of revision. However, I won't object to accepting it if my co-reviewers champion it.

**Justification For Ethical Concerns:**

Since the dataset is not available for the review, it is not decidable for sure whether there is no ethical issue in this work.


**Reasons To Accept:**

1. The proposed MAVERICS-XM3600 dataset should be a valuable asset for multilingual VQA research.

**Reasons To Reject:**

1. Although this work summarizes the main contributions in the last paragraph of section 1, the contributions (ii)-(iv) are rather weak.
(1) The multilingual data curation pipeline is largely based on VQ2A (Changpinyo et al., 2022). It is a somewhat straightforward extension of VQ2A by adding some translation steps (section 3.2).

(2) The benchmark creation process contains little description of novelty.
(3) The MPT model could be a useful off-the-self baseline for the benchmark but bears little novelty.

2. Experimental results are somewhat pointless.
(1) OFA-Large and BLIP2 are SOTA VQA models, but they are only for English not for other languages. It is unclear whether they are good enough baselines for the experiments. One may try some models (OSCAR or mBERT) used in the xGQA paper.
(2) The proposed MPT model is not as good as SOTA models. As shown in Table 5, it is better than OFA, comparable to BLIP2, and worse than PaLI. This paper argues that it is much smaller, but there is no comparison in terms of model sizes and compute.
(3) The analysis experiments in Table 6-8 show somewhat obvious (Table 6, 8) or mixed results in different metrics (Table 7). For example, it is not surprising that the English trained model works well with French but no with other different languages.

3. Some minor weaknesses
(1) The comparison with xGQA starting from line 421 lacks ground. For example, it argues “GQA’s questions are restricted“ but “MaXM’s questions are more general” with no evidence or examples.



**Reproducibility:**

3: Could reproduce the results with some difficulty. The settings of parameters are underspecified or subjectively determined; the training/evaluation data are not widely available.

**Reviewer Confidence:**

4: Quite sure. I tried to check the important points carefully. It's unlikely, though conceivable, that I missed something that should affect my ratings.

---

> ### Author Rebuttal · Authors · 2023-08-28
>
> **[R1] The multilingual data curation pipeline is…a somewhat straightforward extension of VQ2A by adding some translation steps (section 3.2)...The benchmark creation process contains little description of novelty….The MPT model could be a useful off-the-self baseline for the benchmark but bears little novelty.**
>
> At a high level, we think it is desirable that our framework and proposed models are simple and cost effective. In other words, the significance of our contributions lies in effectively achieving our goal of scaling up VQA to non-English languages without sacrificing simplicity while controlling cost.
>
> At the same time, our approach may *look* straightforward in hindsight but in practice it is difficult to *get the detail right*. This has a lot to do breaking down the task into small pieces and suitably assigning machines or humans to each piece (L342-350). For instance, we develop an automatic filtering approach in Section 3.2 to deal with the fact that existing translation models are not optimized for translating short answers (L237-240). We develop a simple answer bias mitigation approach for certain answer types (Section 3.3). Finally, we did have to iterate our annotation protocol and quality checks multiple times to arrive at what may appear to be straightforward in Section 4.3 and Appendix B.
>
> Overall, our experience indicates designing a rigorous annotation protocol for multilingual VQA to balance quality and efficiency is non-trivial, and it is more complex than “adding translation to VQ^2A”.
>
>
> **[R2] Experimental results are somewhat pointless**
>
> We clarify each point below. We hope to convince the reviewer that these experiments were conducted with sound, scientific purposes and will use additional space to make their messages clearer if the paper gets accepted.
>
> **1) OFA-Large and BLIP2 are SOTA VQA models, but they are only for English not for other languages….unclear whether they are good enough baselines for the experiments.**
>
> We put effort into adapting these English VQA models to the multilingual setting to ensure that our baselines are strong.
>
> First, we take a standard multilingual adaptation approach *Translate-Test* (L465-469), making them much stronger as baselines; see Table 8 for results without adaptation where these baselines perform poorly.
>
> In addition, we found that Translate-Train is inferior to Translate-Test as an adaptation approach. For instance, The answer of finetuned BLIP2 to the French question *“Outre les fleurs roses, quelle autre couleur y avait-il dans le jardin?* (*“Besides pink flowers, what other color was there in the garden?”*) is *“pink”* while the correct answer is *“blanc”* (*“white”*) — wrong both in terms of language and semantics. More generally, it is not immediately obvious how to adapt English VQA models with, for example, vocab and tokenizers that overfit the English language. This suggests that the design of these multimodal models would benefit from having multilinguality in mind from the start.
>
> Finally, our adaptation approach in the xGQA setting outperforms approaches in xGQA, including OSCAR and mBERT. See Appendix D.
>
> **The proposed MPT model is not as good as SOTA models. As shown in Table 5, it is better than OFA, comparable to BLIP2, and worse than PaLI. This paper argues that it is much smaller, but there is no comparison in terms of model sizes and compute.**
>
> We are happy to provide such detail, which we hope serves as a context for comparison and clarify L503-510.
> | Model | Parameter count |
> |-------------|:----------:|
> | OFA-Large | 470M |
> | BLIP2 | 11B|
> | Simple MPT | 1.5B |
> | PaLI | 17B |
>
> **The analysis experiments in Table 6-8 show somewhat obvious (Table 6, 8) or mixed results in different metrics (Table 7). For example, it is not surprising that the English trained model works well with French but no with other different languages.**
>
> We argue that these ablation and sensitivity analysis experiments and results convey the thoroughness of our benchmarking results in Table 5. In particular,
>
> Table 6 shows what happens when we deviate from the training data source in Table 5 (language-wise and captions in the case of VQ^2A-based data). First, Simple MPT in Table 5 does not underperform Single-Language baselines, allowing us to have one unified model with strong results but low cost. Second, the result supports the use of manually-curated VQAv2 in Table 5 for training; it does not contain biases from VQ^2A (used to create our benchmark) but works well enough to beat noisy VQ^2A-CC3M.
>
> Table 7 shows what happens when we deviate from our main evaluation metric. Accuracy — what we adopted in Table 5 — is correlated well with other soft metrics, so this metric (combined with our Answer Expansion and Standardization) is useful. We do not see this as a mixed result.
>
> Table 8 shows that, without adaptation, English SOTA VQA models perform poorly, in contrast to strong results of adapted models we report in Table 5.
>
> ***Minor: …The comparison with xGQA starting from line 421 lacks ground…***
>
> Thank you! We will expand this discussion and provide supporting evidence, including the fact that GQA is explicitly designed for “reasoning” (e.g., abstract in [1]) and an analysis (e.g., Section 5.3) in [2], which our approach is based on.
>
> [1] https://arxiv.org/abs/1902.09506
>
> [2] https://arxiv.org/abs/2205.01883

---

### Official Review · Reviewer_PM6i · 2023-08-04

**Soundness:** 3

**Excitement:**

3: Ambivalent: It has merits (e.g., it reports state-of-the-art results, the idea is nice), but there are key weaknesses (e.g., it describes incremental work), and it can significantly benefit from another round of revision. However, I won't object to accepting it if my co-reviewers champion it.

**Paper Topic And Main Contributions:**

The paper introduces a method predicated on translation to generate multilingual Visual Question Answering (VQA) data. Following a rigorous process of human verification and filtration, it provides a benchmarking tool to gauge the multilingual comprehension capabilities of VQA models. Moreover, the paper presents a lightweight, multilingual VQA baseline model, alongside an analysis of the gathered data.

**Questions For The Authors:**

A. Could you elaborate on the advantages of the proposed multilingual dataset over a straightforward mix of single-language datasets? Given the relatively small size of the proposed dataset, its labor-intensive creation process, and its exclusive use as a benchmark, how does it prove superior?

B. If monolingual VQA models, such as OFA or BLIP2, were fine-tuned akin to SimpleMPT of PALI using multilingual data, would this potentially enhance their performance on the mVQA task?

**Reasons To Accept:**

The paper proposed a new benchmark evaluating the mVQA ability of current VQA models. The dataset utilized has undergone thorough human verification to ensure its quality. Furthermore, the paper offers an exhaustive comparison of monolingual and multilingual VQA models, utilizing the newly proposed benchmark as the evaluative standard.

**Reasons To Reject:**

1. The analysis of the dataset and the multilingual context presented in the paper is brief and lacks sufficient detail. A more comprehensive examination, particularly of the VQA data from different languages, would be highly beneficial for the wider community.

2. The benefits of the proposed test dataset, as opposed to a mixed dataset comprising many single-language VQA data, are not well defined or articulated.

3. The test data size is still somewhat constrained, limiting the potential scope and applicability of the findings.

**Reproducibility:**

4: Could mostly reproduce the results, but there may be some variation because of sample variance or minor variations in their interpretation of the protocol or method.

**Reviewer Confidence:**

4: Quite sure. I tried to check the important points carefully. It's unlikely, though conceivable, that I missed something that should affect my ratings.

---

> ### Author Rebuttal · Authors · 2023-08-29
>
> **[R1] The analysis of the dataset and the multilingual context presented in the paper is brief and lacks sufficient detail. A more comprehensive examination, particularly of the VQA data from different languages, would be highly beneficial...**
>
> We are unsure of how our current analysis is brief and lacks sufficient detail. But we would be more than happy to make it more comprehensive, if the reviewer wouldn’t mind clarifying this further. Here’s what is currently available:
>
> Section 4.4 (along with Table 4 and Figure 4) discusses the statistics of our dataset and how our dataset is complementary to/different from xGQA. Appendix B2 further highlights the richness of our dataset.
>
> Appendix B2 also discusses special questions: “Collection” questions, which we included and “Ambiguous” and “RAI-sensitive” questions, which we discarded but believe can serve as guideline for future annotation processes.
>
> Qualitative examples are provided in Figure 1, Figure 2, Figure 8, Figure 7, Figure 9, Figure 10, and Figure 11, with Figure 2 and Figure 8 focusing on the annotation process. These figures include phenomena specific to particular languages (L892-896), for example, Chinese text in Figure 1.
>
> The discussion of language-specific context also appears in Section 4.1 and Figure 3 (why we focus on deriving our benchmark from the XM3600 dataset), L328-333, L533-537, Figure 5 (Instructions which are specific to particular languages based on our analysis), and Appendix B2.
>
> **[R2] The benefits of the proposed test dataset, as opposed to a mixed dataset comprising many single-language VQA data, are not well defined or articulated**
>
> **[QA] Could you elaborate on the advantages of the proposed multilingual dataset over a straightforward mix of single-language datasets? Given the relatively small size of the proposed dataset, its labor-intensive creation process, and its exclusive use as a benchmark, how does it prove superior?**
>
> Existing single-language VQA datasets ***do not exist*** for non-English languages, except for xGQA (L100-110), which focuses on the cross-lingual setting and reasoning (L403-431). Thus, one cannot currently perform *a straightforward mix of single-language datasets*. In theory, a mix of single-language VQA datasets may likely be more biased toward intra-cultural rather than inter-cultural phenomena (e.g., the ability to answer questions in French for images that cover Chinese culture, L296-306).
>
> Further, the standard approach to curating VQA data for a particular language is based almost entirely on human annotators, thus resource-intensive (L46-51). Our approach is superior because it is more automatic, actually providing a much *less labor-intensive* path to the multilingual setting (L51-58).
>
> Finally, establishing an evaluation benchmark is the first step toward measuring progress in multilingual VQA. It is often regarded as the most difficult task in data creation as it requires both the data to be clean and the metric reliable. The requirement is much less strict for training data creation, where much of human annotation steps in Section 4.3 can be relaxed or removed. Its exclusive use as a test-only benchmark follows previous work such as nocaps and Crossmodal-3600, which aims to test the ability of models to generalize (to new concepts or languages, and in our cases to the new image domain as mentioned in L307-314)
>
> **[QB] If monolingual VQA models, such as OFA or BLIP2, were fine-tuned akin to SimpleMPT of PALI using multilingual data, would this potentially enhance their performance on the mVQA task?**
>
> In our early experiments, we found that Translate-Train is inferior to Translate-Test in adapting for these English VQA models. For instance, The answer of finetuned BLIP2 to the French question *“Outre les fleurs roses, quelle autre couleur y avait-il dans le jardin?* (*“Besides pink flowers, what other color was there in the garden?”*) is *“pink”* while the correct answer is *“blanc”* (*“white”*) — wrong both in terms of language and semantics. More generally, it is not immediately obvious how to adapt English VQA models with, for example, vocab and tokenizers that overfit the English language. This suggests that the design of these multimodal models would benefit from having multilinguality in mind from the start.

---

### Official Review · Reviewer_NrNZ · 2023-08-05

**Typos Grammar Style And Presentation Improvements:** N/A
**Soundness:** 4

**Excitement:**

4: Strong: This paper deepens the understanding of some phenomenon or lowers the barriers to an existing research direction.

**Missing References:**

N/A

**Paper Topic And Main Contributions:**

This paper proposes new solutions for multilingual visual question answering (mVQA). Specifically, they contribute to this area by presenting (i) a framework to reduce the cost in terms of manual annotation and (ii) a test benchmark for mVQA called MAVERICS-XM3600 (MaXM) translated in 7 languages.

Their proposed framework can be summarized in four steps. First, they automatically translate a non-English caption to an English one. Second, starting from the translated caption, they generate a set of QA pairs. Third, they translate the QA pairs to the target language, and lastly, they filter the translated pairs removing all the answers not correlated with the original caption.

To evaluate and test this new benchmark (MaXM), they trained a model named “simple MPT” using different data settings, and then they used existing models (e.g., OFA, BLIP2, PaLI) showing the benchmark capabilities.

**Questions For The Authors:**

- Did you notice if this approach generates some artifacts or some wrong pattern?

**Reasons To Accept:**

- They propose contributing to this research area with a new dataset that could help expand mVQA to a multilingual setting.
- They perform an exhausting set of experiments and ablations.
- The paper is well-written and easy to follow.

**Reasons To Reject:**

- I would appreciate an in-depth qualitative analysis of the dataset.

**Reproducibility:**

4: Could mostly reproduce the results, but there may be some variation because of sample variance or minor variations in their interpretation of the protocol or method.

**Reviewer Confidence:**

3: Pretty sure, but there's a chance I missed something. Although I have a good feel for this area in general, I did not carefully check the paper's details, e.g., the math, experimental design, or novelty.

---

> ### Author Rebuttal · Authors · 2023-08-29
>
> **[R] I would appreciate an in-depth qualitative analysis of the dataset.**
>
> We are unsure what *an in-depth qualitative analysis of the dataset* entails, but we would be happy to provide them, if the reviewer wouldn’t mind clarifying this further. Here’s what currently is available:
>
> Section 4.4 (along with Table 4 and Figure 4) discusses the statistics of our dataset and how our dataset is complementary to/different from xGQA. Appendix B2 further highlights the richness of our dataset.
>
> Appendix B2 also discusses special questions: “Collection” questions, which we included and “Ambiguous” and “RAI-sensitive” questions, which we discarded but believe can serve as guideline for future annotation processes.
>
> Qualitative examples are provided in Figure 1, Figure 2, Figure 8, Figure 7, Figure 9, Figure 10, and Figure 11, with Figure 2 and Figure 8 focusing on the annotation process. Figure 3 provides a qualitative illustration of why we focus on deriving our benchmark from the XM3600 dataset.
>
> **[Q] Did you notice if this approach generates some artifacts or some wrong pattern?**
>
> Most artifacts come from translation, which we discuss and mitigate in Section 4.2. Figure 2 and Figure 8 provide examples of how we correct/remove these artifacts.

---

### Meta-Review · Area_Chair_AXYo · 2023-09-11

**Recommendation:** 3

**Metareview:**

The paper creates a dataset MaXM which is a multilingual question answering benchmark. The dataset serves as a test-only mVQA dataset to validate the mVQA model performance. The reviewers have concern on the technical contribution from this paper and also request some detailed analysis on the proposed dataset. But overall, this dataset would be useful to the community as an evaluation set. The author has promised a release of this dataset in the paper.

---

### Decision · Program_Chairs · 2023-10-07

**Decision:**

Accept-Findings

**Comment:**

The paper creates a dataset MaXM which is a multilingual question answering benchmark. The dataset serves as a test-only mVQA dataset to validate the mVQA model performance. The reviewers have concern on the technical contribution from this paper and also request some detailed analysis on the proposed dataset. But overall, this dataset would be useful to the community as an evaluation set. The author has promised a release of this dataset in the paper.